# Design of a Low-Cost Flat E-Band Down-Converter with Variable Conversion Gain

**DOI:** 10.3390/s25175492

**Published:** 2025-09-03

**Authors:** Mehrdad Harifi-Mood, Mansoor Dashti Ardakani, Djilali Hammou, Emilia Moldovan, Bryan Hosein, Serioja O. Tatu

**Affiliations:** 1Place Bonaventure, Centre-Énergie, Matériaux et Télécommunications, Institut national de la recherche scientifique, University of Quebec (INRS), Montreal, QC H5A 1K6, Canada; mansoor.dashti@inrs.ca (M.D.A.); djilali.hammou@inrs.ca (D.H.); emilia.moldovan@inrs.ca (E.M.); serioja.ovidiu.tatu@inrs.ca (S.O.T.); 2Focus Microwaves Inc., Saint-Laurent, QC H4S 1A8, Canada; bryan@focus-microwaves.com

**Keywords:** millimeter-wave, mixer, MHMIC, ultra-wideband, maximally flat gain

## Abstract

This paper presents the design and implementation of a wideband diode-based down-converter operating from 60 to 90 GHz with a variable flat conversion gain. The proposed down-converter is implemented utilizing the Miniature Hybrid-Microwave Integrated Circuit (MHMIC) technology. It is composed of a wideband double-balanced mixer, a Local Oscillator (LO) chain, and a differential TransImpedance Amplifier (TIA) with a variable gain. The designed mixer uses a novel topology exhibiting minimum reflection and high isolation between the RF and LO ports across a wide operating frequency of 30 GHz. In this topology, two balanced detectors generate the differential IF signal with minimum reflection. The characteristic impedance ($Z_{0}$) of the mixer is set to be 70.7
Ω, to minimize trace widths to reduce the mutual coupling and increasing the bandwidth. The OPA 657 is the core of the designed differential TIA with a variable gain. In addition, the LO chain of the down-converter utilized a combination of an active (×2) and a passive (×3) multiplier to generate enough RF power in the desired frequency range. Also, a WR-12 waveguide to Substrate Integrated Waveguide (SIW) transition is designed for the RF and LO ports that operates through the E-band. The proposed down-converter demonstrates excellent performance, with a high isolation between RF and LO ports exceeding 22 dB and a maximum conversion gain of −5 dB, and a response with a variation of ±5 dB across the band. The proposed mixer exhibits a return loss of better than 10 dB at both RF and LO ports, and it consumes a power of 560 mW.

## 1. Introduction

The constant and rapid evolution of wireless communication systems [1,2,3], radar technologies [4,5,6,7,8], and sensing applications [9,10,11,12] has driven the demand for high-performance, wideband transceivers operating in the millimeter-wave (mm-wave) and sub-terahertz frequency ranges. Among the critical components in RF transceiver design, mixers play a pivotal role in frequency conversion, which enables exploiting higher bandwidths available in higher-frequency ranges. As the operating frequencies of telecommunication systems continue to increase, the design of cost-effective mixers capable of exhibiting low conversion loss and high linearity throughout a wide bandwidth has become a significant challenge. Mixers can be designed and fabricated using different technologies such as CMOS [13,14,15], GaAs [16,17,18], InP [19,20,21,22,23], etc. Each technology offers distinct advantages, such as higher integration or better high-frequency performance. Despite some advantages that these technologies may have, they are generally less cost-efficient for small-scale fabrications.

On the other hand, diode base mixers can exhibit good performance in higher frequencies, and they are more cost-effective. In this context, Schottky diodes are inherently nonlinear devices that are good candidates for designing higher-frequency mixers, primarily due to their low junction capacitance. This characteristic minimizes parasitic capacitance, enabling high-speed operation critical for mm-wave frequencies. Additionally, their low threshold voltage reduces the LO power required to drive the mixer, which can improve efficiency and simplify system design with lower price [24]. Planar multi-port circuits on thin-film ceramic can be considered as a promising candidate, particularly due to their low-cost and low-complexity designs [25]. Although by considering the minimum thickness of ceramic in the market, this solution presents significant challenges, mostly in higher frequency ranges (>60 GHz), which must be addressed. For instance, one primary issue is the high insertion loss of the transmission lines due to the roughness of the substrates. Furthermore, given the frequency-dependent variation of wavelength, creative circuit structures are necessary to effectively broaden the bandwidth.

This paper presents the design and performance analysis of an E-band down-converter implemented using Miniature Hybrid-Microwave Integrated Circuit (MHMIC) technology. The mixer of the down-converter employs GaAs Schottky diodes (HSCH-9161), which are well suited for mm-wave applications due to their low junction capacitance and high cut-off frequency. One of the primary features of the proposed down-converter mixer is its cost-effectiveness for high-performance, small-scale, or prototype fabrications compared to alternative technologies, such as Monolithic Microwave Integrated Circuits (MMICs) or mixers based on semiconductor processes as SiGe or InP. Furthermore, the proposed topology of the mixer shows a flat response across a wide frequency range of the E-band. This cost advantage, combined with the wide operational bandwidth, makes it an attractive candidate for next-generation communication and sensing systems.

A novel architecture incorporating a parabolic open stub is introduced and suited after the diodes with a miniaturized size. The proposed open stub effectively rejects the RF and LO signals within the frequency range of 60 to 90 GHz with a small occupied area. This parabolic open-stub architecture can be used for different purposes, such as wideband efficiency improvement of power amplifiers, bias networks, RF grounding, etc.

The LO power to drive the mixer of the down-converter is generated using a two-stage LO chain. The designed LO chain uses an active GaAs ×2 module (HMC578) from Analog Devices as the first stage, plus a passive GaAs ×3 GaAs-based module (HMC-XTB110) from Analog Devices. These modules are integrated with two pass-band filters to suppress any unwanted harmonics, making a ×6 LO chain for the down-converter.

In addition to the mixer design, the proposed down-converter utilizes a differential TransImpedance Amplifier (TIA) with variable gain at the output stage. This integration not only amplifies the down-converted signal but also provides flexibility in adjusting the gain to suit specific application requirements. The OPA 657 chip from Texas Instruments Inc., Dallas, TX, USA is used to design this TIA.

Figure 1 depicts the proposed down converter in a receiver using an antenna and an LNA in the RF path. The combination of wide bandwidth, low, variable, and flat conversion loss makes this down-converter a good candidate for mm-wave applications. Its low-cost MHMIC implementation further enhances its appeal for mass production and deployment in commercial systems. This article provides a comprehensive analysis of the proposed down-converter design and performance. The results demonstrate the potential of this down-converter to address the growing demands of modern RF systems, paving the way for its adoption in a wide range of applications, including 5G/6G communications, automotive radar, and high-resolution sensing.

In the following sections, first, the theory and design of each component of the down-converter are explained. Then, in the Measurement Results and Discussion section, the performance of the down-converter is analyzed and compared with theory and simulation results. Lastly, the Conclusion section summarizes the design, performance, and analysis of the proposed down-converter.

## 2. Theory and Design

Mixers are essential components in transceivers used for frequency conversion, allowing signals to be shifted between a higher RF and a lower IF or baseband. Up-conversion mixers are commonly used in transmitters to enable efficient signal propagation [26]. On the receiver side, mixers down-convert the incoming high-frequency signals to the baseband range. This procedure allows analog-to-digital converters (ADCs) to process the signals effectively. Also, the frequency conversion improves signal selectivity, mainly due to the ease of filter design in lower frequencies. It also simplifies the design of high-gain amplification stages (i.e., baseband amplifier) [27]. The core of mixer design generally evolves around the non-linearity of at least one component [28,29]. Nevertheless, mixers are linear devices in a range of operating powers [30]. The frequency conversion (up or down) can be implemented in single [31,32] or multiple stages, each offering distinct advantages.

One of the main challenges of designing diode-based wideband mixers is the design of the matching networks for the utilized diodes. Since in the design of matching networks the operating bandwidth reduces by increasing the reflection coefficient of the used diodes [33]. The proposed double-balanced mixer uses two wideband $90^{\circ }$ hybrid couplers to alleviate the reflection coefficient of the circuit caused by the high inherent reflection of the diodes. Figure 2 shows the schematic of the mixer. Considering identical reflection from each diode, the reflected energy from the diodes are being absorbed by the positioned loads at port 4 of these two couplers [34]. Therefore, the circuit represents a minimal reflection at its input ports. In this design, a planar double-balanced architecture is used to achieve high linearity with higher conversion gain. In addition, the planar architecture reduces the fabrication process complexity. As can be seen, this circuit is composed of three $90^{\circ }$ hybrid couplers, a $90^{\circ }$ phase shifter, and the amplification stages.

To analyze this circuit, two signals of LO and RF with frequencies of $\omega_{LO}$, and $\omega_{RF}$, respectively, and a phase of $0^{\circ }$ are considered to be available at ports 1 and 4 of coupler 1. Consequently, the incident waves to couplers 2 and 3 can be given as below:(1a)$$a_{1(coupler2)}=\frac{1}{\sqrt{2}}\left(-ja_{1}e^{j\omega_{LO}t}-a_{4}e^{j\;\omega_{RF}t}\right)$$(1b)$$a_{1(coupler3)}=\frac{1}{\sqrt{2}}\left(ja_{1}e^{j\omega_{LO}t}-a_{4}e^{j\;\omega_{RF}t}\right)$$ Assuming $a_{1}=V_{LO}/\sqrt{Z_{0}}\times cos\left(\omega_{LO}t\right)$, $a_{4}=V_{RF}/\sqrt{Z_{0}}\times cos\left(\omega_{RF}t\right)$ and $v_{diode}=\sqrt{Z_{0}}\times a_{coupler}$, the voltages across diodes 1 and 3 are given as below:(2a)$$\begin{array}{cc} v_{D1} & =\sqrt{\frac{Z_{0}}{2}}\left(V_{LO}\cos\left(\omega_{L0}t-\frac{\pi }{2}\right)-V_{RF}\cos\left(\omega_{RF}t\right)\right)\\ & =\sqrt{\frac{Z_{0}}{2}}\left(V_{LO}\sin\left(\omega_{L0}t\right)-V_{RF}\cos\left(\omega_{RF}t\right)\right) \end{array}$$(2b)$$\begin{array}{cc} v_{D3} & =\sqrt{\frac{Z_{0}}{2}}\left(V_{LO}\cos\left(\omega_{L0}t+\frac{\pi }{2}\right)-V_{RF}\cos\left(\omega_{RF}t\right)\right)\\ & =-\sqrt{\frac{Z_{0}}{2}}\left(V_{LO}\sin\left(\omega_{L0}t\right)+V_{RF}\cos\left(\omega_{RF}t\right)\right) \end{array}$$ Using the Taylor expansion approximation, the I−V characteristic of the non-linear diode can be expressed as:(3)$$i\left(v\right)=k_{0}+k_{1}v+k_{2}v^{2}+k_{3}v^{3}+k_{4}v^{4}+\ldots$$ Therefore, the current through each of the diodes can be calculated as follows:(4a)$$\begin{array}{c} i_{D1}=k_{0}+k_{1}\sqrt{\frac{Z_{0}}{2}}\left(V_{LO}\sin\left(\omega_{L0}t\right)-V_{RF}\cos\left(\omega_{RF}t\right)\right)\\ +k_{2}\frac{Z_{0}}{2}{\left(V_{LO}\sin\left(\omega_{L0}t\right)-V_{RF}\cos\left(\omega_{RF}t\right)\right)}^{2}+\ldots \end{array}$$(4b)$$\begin{array}{c} i_{D3}=k_{0}+k_{1}\sqrt{\frac{Z_{0}}{2}}\left(V_{LO}\sin\left(\omega_{L0}t\right)+V_{RF}\cos\left(\omega_{RF}t\right)\right)\\ +k_{2}\frac{Z_{0}}{4}{\left(V_{LO}\sin\left(\omega_{L0}t\right)+V_{RF}\cos\left(\omega_{RF}t\right)\right)}^{2}+\ldots \end{array}$$ By expanding the second-order term, the diodes current can be rewritten as:(5a)iD1=⋯+k2Z04[VLO221−cos(2ωLOt)+VRF221+cos(2ωRFt)−VLOVRFsin((ωLO+ωRF)t)−VLOVRFsin((ωLO−ωRF)t)]+…VLO22(5b)iD3=⋯+k2Z04[VLO221−cos(2ωLOt)+VRF221+cos(2ωRFt)+VLOVRFsin((ωLO+ωRF)t)+VLOVRFsin((ωLO−ωRF)t)]+…VLO22 Higher-order terms are being suppressed by the open stubs, although lower frequencies that are generated by the even-order terms (i.e., “$k_{2n}{\left(Z_{0}/2\right)}^{2n-1}({\left(V_{LO}sin\left(\omega_{LO}t\right)+V_{RF}cos\left(\omega_{RF}t\right)\right)}^{2n}$”) can pass through the TIA. For instance, the 4th-order term generates a signal at frequency of $2\omega_{LO}-2\omega_{RF}$, which is equal to $2\omega_{IF}$. Keeping the IF frequency constant can make the suppression of these harmonics feasible by limiting the bandwidth of the TIA.

By using a differential TIA for the proposed mixer, the output signal can be calculated as follows:(6)$$V_{IF}=G_{TIA}\times \left(I_{IF(A)}-I_{IF(B)}\right)=G_{TIA}\times 2\left[\frac{k_{2}Z_{0}}{4}V_{RF}V_{LO}sin\left(\omega_{LO}-\omega_{RF}\right)\right]$$ Therefore, the DC terms of the diodes 1 and 3 are canceled, and the generated IF signals are doubled. Consequently, the required LO power to drive the mixer is reduced significantly. In these calculations, the reflection coefficients of the diodes and the couplers are considered to be zero, while Schottky diodes typically represent a highly reflective behavior. So, assuming a reflection coefficient of $\Gamma_{D}$ for each of the diodes, the reflected waves at ports 1 and 4 of coupler 2 (i.e., $b_{1}$ and $b_{4}$) can be given as follows:(7a)$$b_{1}=\frac{a_{1}}{2}\left(\Gamma_{D2}-\Gamma_{D1}\right)$$(7b)$$b_{4}=\frac{ja_{1}}{2}\left(\Gamma_{D1}+\Gamma_{D2}\right)$$
where $\Gamma_{D1}$ and $\Gamma_{D2}$ are reflection coefficients of the diodes 1 and 2, respectively. Similarly, the reflected waves at ports 1 and 4 of coupler 3 can be calculated. As can be revealed from Equation ([Disp-formula FD7a-sensors-25-05492]), by utilizing identical diodes (i.e., $\Gamma_{D1}=\Gamma_{D2}=\cdots$), the reflected waves of the couplers is being equal to 0 ($b_{1}=0$). Hence, despite using highly reflective diodes, the proposed mixer can exhibit a low return loss in a wide operating band thanks to the presence of diodes 2 and 4. Although in practice, the design of the proposed wideband mixer in the mm-wave frequency range has various challenges that need to be addressed.

### 2.1. WR-12 to Microstrip Transition

Generally, one of the main challenges in the design of mm-wave circuits and components is energy transfer. Since the high transmission loss in this frequency range reduces the efficiency significantly. This issue becomes even more problematic by broadening the operating bandwidth of the designed circuit, mainly because the thermal noise floor will be increased, and it is easier for the signal to be drowned into noise, making it obscured.

One of the low-transmission-loss solutions for transferring energy is the waveguide, which was introduced in 1897 [35] and rediscovered 40 years later by G. C. Southworth and W. L. Barrow separately [36]. As mentioned, transferring energy through waveguides in their operating frequency band has very low transmission loss. So, to overcome the high transmission loss in the operating frequency of the circuit, E-Band, the proposed circuit is designed to operate using a standard WR-12 waveguide. However, to excite the proposed planar mixer with LO and RF signals, a waveguide to microstrip transition is designed and simulated in HFSS.

The designed transition used substrate-integrated waveguide (SIW) [37]. A ceramic substrate with a thickness of 127 μm and a permittivity of 9.9 is used to minimize the insertion loss through the desired frequency band. Figure 3 depicts the proposed transition optimized to exhibit the minimum reflection and transition loss over the frequency band of 60 to 90 GHz.

Simulation results of the proposed transition are shown in Figure 4. As can be seen, the E-band WR-12 to microstrip transition shows an insertion loss of less than 1 dB, and a return loss of better than 12 dB throughout the operating frequency band. These simulation results are matched with measurement results of the back-to-back transition investigated in [38].

### 2.2. 90° Hybrid Coupler

The proposed architecture utilizes three $90^{\circ }$ hybrid couplers to combine the RF and LO signals and deliver them to the Schottky diodes with the desired phases for the optimized down-conversion process. Basically, an ideal $90^{\circ }$ hybrid coupler represents high isolation between ports 1 and 4, and combined powers from these ports will be delivered to ports 2 and 3 with $90^{\circ }$-phase difference. To widen the bandwidth of the hybrid coupler, a ring-shaped design is considered. As is known, sharp corners introduce parasitic capacitance, causing a narrower operating band. Moreover, mutual coupling between the traces that are close to each other increases significantly by increasing the frequency. This phenomena reduce the bandwidth of the coupler significantly. So, to minimize this effect, the coupler needs to be designed with 3λ/4 or 5λ/4 trace lengths instead of λ/4, which reduces the bandwidth in another way (i.e., the phase difference between the beginning and the end frequencies of the band would be very different). Another method used in this design is reducing the width of the traces, which can be reduced down to the fabrication resolution, which means that the characteristic impedance of the couple would be increased. The impedance of the coupler can be transformed using a simple quarter-wavelength taper to be matched with the other components of the circuit. Due to utilizing a thin substrate with high permittivity, the quarter-wavelength taper can exhibit a wide operating bandwidth with minimal impact on the circuit bandwidth. Considering the resolution of the utilized fabrication process (i.e., 50 μm in this case), the designed coupler has a characteristic impedance of 70.7
Ω.

The designed hybrid coupler uses a symmetry architecture, and it can be assumed that:(8a)$$S_{12}=S_{34}$$(8b)$$S_{13}=S_{24}$$(8c)$$S_{14}=S_{23}$$ Consequently, to evaluate the designed coupler, three circuits need to be fabricated. Figure 5 illustrates two microphotographs of the circuit designed for two coupling paths measurements and the phase difference between them. These two circuits are designed for on-wafer measurement with Ground–Signal–Ground (GSG) probes with a pitch size of 150 μm. Therefore, two RF grounds are located at the ground pads of the GSG probes. In addition, two unused ports for these measurements are terminated with 70.7
Ω loads in order to perform accurate two-port measurements.

Two-port measurement results of the fabricated coupler are shown in Figure 6. The return loss of the circuit from all the ports is better than 10 dB. Plus, the power is equally divided into two parts, hence, the $S_{12}$ and $S_{13}$ of the circuit are around −3 dB across the entire frequency band of 30 GHz. In addition, the isolation of the coupler is better than 20 dB throughout the operating frequency band.

### 2.3. Parabolic Open Stub

Open stubs can demonstrate high suppression for undesired frequencies in planar structures. However, as the length of the open stub should be equal to the quarter wavelength of the intended frequency in the substrate, it can cover a relatively narrow frequency band. To cover a wider frequency band, designers usually cascade several open stubs, which will occupy more area. Cascading several open stubs can cause a high insertion loss for a wider frequency range, but not necessarily a high return loss, because by cascading each single open stub, which is designed for a specific frequency, the length of the open-stub chain increases, inevitably. A higher length exhibits higher insertion loss, especially in higher frequencies (i.e., considering the employed substrate). Consequently, the returned waves from each open stub would be weaker in proportion to the distance of the open stub and the intended reflection point. High reflection throughout a wide frequency band can be pivotal for some applications, such as designing efficient power amplifiers, mixers, etc. On the other hand, cascading several open stubs occupies a larger area, which leads to higher cost.

Parabolic open stub, which is proposed and used in this mixer design, can make a high reflection coefficient throughout a wide frequency band with a compact size. The parabolic open stub, unlike radial open stubs, has a variable radius to cover a higher frequency band. Basically, this scheme consists of an infinite number of conventional open stubs that their lengths are changing continuously, with the size of a radial open stub. The longest and shortest radii, respectively, are proportional to the quarter-wavelength of the lower and higher frequencies of the intended band. Figure 7 depicts the layout of the proposed parabolic open stub and its equivalent circuit and layout with rectangular open stubs.

As the name suggests, the proposed open stub is using a parabolic arc where its minimum and maximum distances to the center of the closed shape correspond to a quarter-wavelength of the maximum and minimum frequencies of the operating band, respectively. It is shown in Figure 7c that the equivalent circuit of the open stub consists of an infinite number of series-LC networks with distinct resonance frequencies ($f_{0}$) within the operating frequency band. Hence, a low impedance is represented at the input of the open stub ($Z_{in}$) for a wide range of frequencies (i.e., wideband high reflection) [39]. The radius of the open stub as a function of the central angle can be given as follows:(9)$$r_{\left(\theta \right)}=\frac{1}{4}\left(\lambda_{H}+\theta .\left(\frac{\lambda_{L}-\lambda_{H}}{\alpha }\right)\right)$$
where $r_{\theta }$ is the radius in mm as a function of the starting angle; $\lambda_{H}$ and $\lambda_{L}$ are wavelengths of the highest and lowest frequency in the band, respectively. Also, α denotes the maximum angle of the sector.

Calculating the occupied area by the parabolic open stub, the total equivalent capacitance of the proposed open stub can be derived as follows:(10)$$C_{tot}=\varepsilon_{r}.\frac{-2.12{\left(\frac{\lambda_{H}}{4}\right)}^{3}-0.5{\left(\frac{\lambda_{H}}{4}\right)}^{2}+\sqrt{2}\left(\frac{\lambda_{H}}{4}\right).\left(\frac{\lambda_{L}}{4}\right)}{H}$$
where $\varepsilon_{r}$ and *H*, respectively denote the dielectric constant and thickness of the utilized substrate.

#### Proof of Concept

To evaluate the performance of the proposed open stub with the minimized size, it needs to show the maximum reflection over the wide frequency band of 60 to 90 GHz. In other words, the ideal open stub needs to exhibit a good RF grounding in the desired frequency band. So, a resistor with the value of $Z_{0}$ can be grounded and show the minimum reflection in a certain bandwidth. An ideal termination needs to exhibit minimum reflection through its operating bandwidth. In other words, the energy needs to be absorbed by the resistor and the RF ground without any reflection. Figure 8 shows the current distribution of three 50-Ω loads that are grounded via three different open rectangular, radial, and parabolic stubs. These open stubs are designed to show an optimized performance throughout the frequency range of 60 to 90 GHz. It can be seen that the rectangular and radial open stubs show higher reflection over the frequency band of 60 to 90 GHz. In addition, it is obvious that a higher portion of the energy is reflected by the rectangular and radial compared to the parabolic open stub over the operating bandwidth.

To compare the performance of the proposed parabolic open stub, three different layouts were implemented on a 127 μm thin-film ceramic substrate using MHMIC technology. Figure 9 shows the fabricated terminations using different open stubs. Similar to the designed $90^{\circ }$ coupler measurement layouts, GSG pads are used with a pitch size of 150 μm. The measurement results of the designed termination using conventional (rectangular), radial, and parabolic open stubs are shown in Figure 10. These 1-port measurements are performed over the frequency bandwidth of 60 to 90 GHz. Bearing in mind that the negligible minimum mismatch is due to the fabrication error in implementing the size of the used resistor.

### 2.4. LO Chain

An RF multiplier is a nonlinear circuit used to increase the frequency of an input signal by an integer factor, making it a critical component in systems requiring higher-frequency signals. Similar to mixer design, frequency multipliers operate by exploiting the nonlinearity of devices like diodes or transistors, which generate harmonics when driven by an RF signal.

The LO chain for this mixer is designed to deliver a relatively flat power throughout the desired band. The first stage of the LO chain is the active GaAs MMIC of HMC 578 from Analog Devices. This module has a multiplication coefficient of 2 alongside an amplifier. This module has an operating input frequency range of 10 to 15 GHz and can deliver a flat power in the frequency range of 20 to 30 GHz. This active multiplier operates with a DC power consumption of approximately 500 mW and is capable of delivering an average output power of +20 dBm when driven by an input signal of +5 dBm. Following this stage, a band-pass filter is designed on a 254 μm thin-film ceramic substrate using the HMIMIC technology to suppress the unwanted harmonics. Then, the signal is fed into the HMC-XTB110, a passive GaAs frequency multiplier designed to increase its input frequencies by a factor of 3. The input frequency of this module is from 20 to 30 GHz with a power of 19 dBm. So, the output of this module is a signal with a frequency range of 60 to 90 GHz with a relatively flat power of 0 dBm. This passive stage introduces a conversion loss of approximately 20 dB. Then another band-pass filter is designed on a 127 μm ceramic substrate to suppress the undesired harmonics. A thinner ceramic substrate is utilized for this design to minimize the insertion loss of the designed filter [40]. Then, the output power in the frequency range of 60 to 90 GHz is delivered to a standard waveguide of WR-12 through the microstrip to WR-12 transition, which was analyzed before.

Figure 11 shows the designed E-band LO chain. To evaluate this design, an RF power of +5 dBm in the operating frequency range of 10 to 15 GHz is injected to the input SMA connector of the circuit. To this aim, the E8254A Agilent signal generator is used. The mm-wave output signal is also measured by the N9040B-UXA Keysight signal analyzer alongside the smart external waveguide mixer of M1971E (Keysight Technologies, Inc., Santa Rosa, CA, USA).

Figure 12 shows the measurement setup used for evaluating the performance of the designed E-band LO chain. The measurement results of the designed E-band LO chain are shown in Figure 13. As can be seen, the output power has a variation of ±3 dB through the E-band output frequency.

### 2.5. Variable-Gain TIA

Typically, TIAs exhibit high input impedance and 50-Ω output impedance. The designed mixer exhibits high output impedance due to the used Schottky diodes. However, the standard characteristic impedance of the cables and measurement instruments is 50 Ω. Therefore, a TIA is designed for the proposed mixer.

Figure 14 shows the schematic of the designed TIA. The IF amplification stage utilizes the Texas Instruments Inc. “OPA 657” operational amplifier, which amplifies the IF signal generated by the mixer. The fixed IF signal at the frequency of 16 MHz carries the down-converted information from the mm-wave frequencies of the RF input and requires amplification to ensure optimal signal level for further processing. As the double-balanced mixer delivers a differential signal to the output, a differential TIA is designed to enhance the conversion gain of the mixer. The designed TIA is showing an input impedance of 5 KΩ with an output impedance of 50 Ω. The two-stage configuration increases the gain of the TIA to a maximum voltage gain of 100 (40 dB). The variable-gain feature, implemented using a variable resistor in the feedback network, allows for adjustable amplification of the 16 MHz IF signal, enabling fine-tuning to accommodate varying signal strengths and system requirements. The 3-dB bandwidth of this circuit is 20 MHz, and it consumes a power of 60 mW approximately.

## 3. Measurement Results and Discussion

As shown in Figure 2, the proposed mixer consists of three $90^{\circ }$ hybrid couplers, a $90^{\circ }$ phase shifter, four Schottky diodes, and the IF amplification stage. To minimize the insertion loss, the planar mixer is designed on a 127 μm ceramic substrate using the MHMIC technology. The permittivity of the utilized substrate is 9.9 with a 1 μm of gold layer as the conductor material. In addition, the $TiO_{2}$ resistive layer has a thickness of 20 nm with a resistivity of 100 Ω/sq under the gold layer. The resistive layer increases the durability of the gold layer from peeling off the smooth surface of the ceramic as well.

Figure 15 shows a microphotograph of the designed mixer. In this design, two WR-12 to microstrip transitions are used to keep the insertion losses for the mm-wave signals of LO and RF as low as possible. Moreover, the well-known zero-bias HSCH-9161 Schottky diode is used as the nonlinear component of the mixer in a planar double-balanced structure. The characteristic impedance of the circuit is designed to be 70.7
Ω to widen the operating bandwidths of the couplers. Therefore, two tapers are used to connect the circuit to the 50 Ω transitions. Proposed parabolic open stubs are used as the reflectors after the Schottky diodes to have a high reflection for the frequency band of 60 to 90 GHz with a minimized occupied area. The two outputs of the mixer deliver the downconverted signal with a phase difference of $180^{\circ }$ to the designed differential TIA. The proposed circuit can exhibit the maximum efficiency considering identical reflection from the used diodes, which has been studied in [34].

Equation (5) reveals that the IF output signals at ports A and B of the mixer have a $180^{\circ }$ phase difference. The designed $90^{\circ }$ phase shifter can cause a phase error at the output of the mixer across the E-Band frequency range. Although due to the high permittivity of the used substrate, this phase error is negligible. To analyze the functionality of the proposed down-converter, the voltage waveforms at ports A and B of the mixer are measured using the Tektronix MSO70804C oscilloscope.

Figure 16 shows the normalized voltage waveform of the mixer at ports A and B in three different RF frequencies of 60, 75, and 90 GHz, while the IF frequency is set to be at 16 MHz. As can be seen in Figure 16, the output voltages exhibit a phase difference of $180^{\circ }$ at the central frequency with an approximate phase error of $10^{\circ }$ at lower and higher frequencies of the operating frequency band. This phase error causes a negligible conversion gain drop of ≃0.13 dB approximately.

Figure 17 depicts the measured IF spectrum of the mixer versus the simulation results. As discussed, even-order terms in Equation (5) generate spurious in the range of IF. The spurious response of the mixer using the HSCH-9161 Schottky diode is simulated in ADS. Due to the limited bandwidth of the TIA, the measured IF harmonics are suppressed to some extent. Furthermore, the spurious response of the designed mixer is simulated using ADS, which is shown in Figure 18. The measured spurious results show up to 20 dB suppression of the unwanted harmonics by the designed TIA through the operating frequency band. For this evaluation, RF and LO power levels are set at −5 dBm using N5295AX53 frequency extenders, and the IF frequency is set to be at 16 MHz.

To evaluate the RF/LO isolation, a waveguide two-port measurement is performed across the operating frequency band. This test used the N5242B VNA alongside with N5292A Millimeter Test Set and N5295AX53 frequency extenders from Keysight Inc. The two-port measurement results of the designed mixer are illustrated in Figure 19.

As can be seen, the proposed mixer is showing a return loss of better than 10 dB across the entire bandwidth of E-band for both ports of LO and RF. Furthermore, the circuit is exhibiting negligible leakage between these two ports. The isolation between these two ports is higher than 22 dB throughout the operating frequency band of 60 to 90 GHz.

The linearity of the designed mixer is evaluated by sweeping the RF power from −50 to 0 dBm in a fixed LO power of −5, 0, and +5 dBm at the center frequency. Figure 20 depicts the conversion gain of the mixer in different RF and LO powers at 75 GHz. It can be seen that the mixer gets compressed in lower RF power if the LO power is higher, which is aligned with theoretical calculations in Equation (Equation 6). N9040B-UXA Keysight signal analyzer is utilized to measure the downconverted signal for this test.

The measured P-1 dB of the mixer at the frequency of 75 GHz in different LO powers is shown in Figure 21. Obviously, stronger LO powers compress the mixer in lower RF power levels. The high linearity of the proposed mixer is mainly due to its double-balanced structure. Basically, the power is divided between four Schottky diodes, making it a suitable candidate for high-power applications in this frequency range.

Figure 22 illustrates the measured normalized conversion gain versus IF frequency. The performance of the mixer in different IF frequencies is also evaluated by sweeping the RF frequency from 75.001 GHz to 80 GHz by a fixed LO power at 75 GHz. As the designed TIA has a limited 3-dB bandwidth of 20 MHz, this test is performed without using the TIA. The IF output signal is measured using a 50-Ω N9040B-UXA Keysight signal analyzer. This measurement shows the potential of the mixer to down-convert signals to an IF from a few Hz to several GHz. For each specific application in which such an interferometric mixer is used, a different TIA needs to be designed for the operating IF frequency with the required bandwidth and gain. For instance, the designed down-converter can be used in a superheterodyne mm-wave receiver for high data-rate applications. It makes receiving several Gb/s communication signals in higher IF frequencies possible.

The key characteristic of a down-converter is the conversion gain at different RF frequencies. So, a wideband down-converter needs to deliver a relatively flat IF output power across its operating frequency band. To evaluate the performance of the deigned down-converter in terms of output power flatness, an RF signal with a power level of −5 dBm is injected into the RF port of the mixer. The LO signal is drawn from the designed ×6 LO chain. In this test, to keep a fixed IF signal at 16 MHz, the LO signal is swept accordingly.

Figure 23 shows the measured IF power versus RF frequency. Considering the fixed LO power of 0 dBm, the measured IF power is showing a conversion gain of −5 dB with a variation of ±5 dB throughout the extensive bandwidth of 30 GHz. The measurement results are partially affected by the generated LO power drawn from the designed LO chain shown in Figure 13. The level of the IF signal can be modified by the variable-gain TIA within a 40 dB window.

Table 1 summarizes and compares the performance of the proposed down-converter with previously reported works in the mm-wave frequency range. The proposed design simultaneously demonstrates high linearity, variable conversion gain, and wide operating bandwidth that covers the entire E-band, while maintaining relatively low fabrication cost. Furthermore, a key advantage of this work is its ability to operate with low LO signal levels, which further reduces both the overall cost and complexity of the circuit.

## 4. Conclusions

In this paper, a wideband double-balanced mixer is designed with a variable conversion gain. Due to the double-balanced structure of the mixer, it shows a high linearity as well. Over the wide frequency band of 60 to 90 GHz, the designed mixer exhibits an output power with a flatness of ±5 dB. The proposed mixer is designed to operate with a standard WR-12 waveguide to minimize the insertion loss of mm-wave signals at ports RF and LO. MHMIC technology is utilized to fabricate the proposed mixer, which makes the designed mixer cost-effective. A thin-film ceramic substrate with a thickness of 127 μm is utilized for this design. The permittivity of the used substrate is 9.9, with a 20 nm $TiO_{2}$, and 1 μm of resistive and metalization layers. This design utilizes a parabolic open stub as the RF reflectors, which represent high reflection across the operating band with reduced size. A ring-shaped $90^{\circ }$ hybrid coupler is designed for this mixer with a characteristic impedance of 70.7
Ω to widen the operating bandwidth. The LO power of the proposed mixer is drawn from the ×6 LO chain. A high isolation of 30 dB is measured between the RF and LO ports of the mixer. Measured return loss of the RF and LO ports are higher than 10 dB throughout the entire E-band. The output IF power level of the mixer is adjustable using a differential TIA within a range of 40 dB. Overall, the proposed mixer with the designed LO chain is a good candidate for wideband applications in mm-wave frequencies with low cost and minimum fabrication complexity.

## Figures and Tables

**Figure 1 sensors-25-05492-f001:**
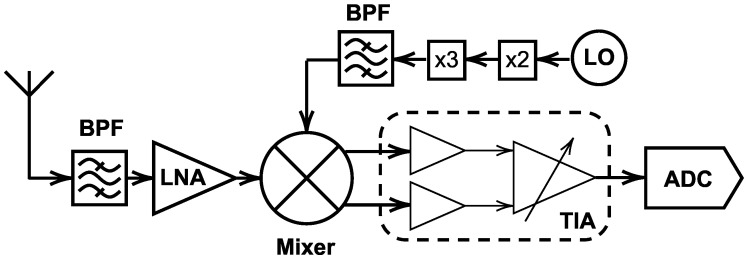
The schematic of the proposed E-band down-converter in a receiver.

**Figure 2 sensors-25-05492-f002:**
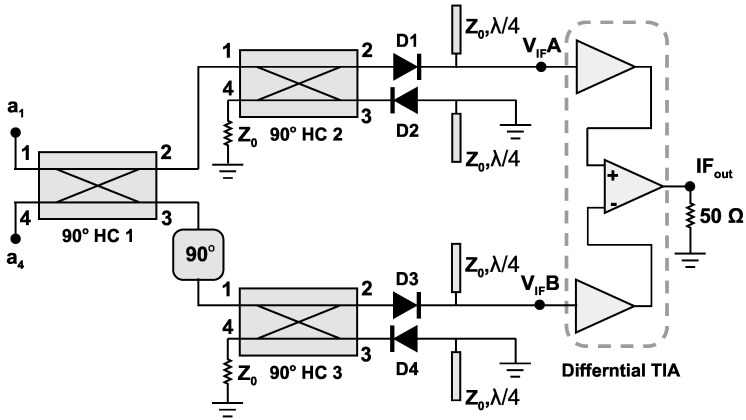
The schematic of the proposed planar double-balanced mixer.

**Figure 3 sensors-25-05492-f003:**
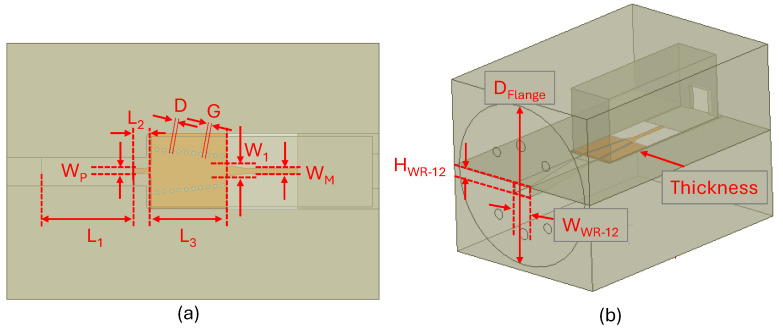
(**a**) Top and (**b**) 3D view of the designed WR-12 to microstrip transition (*$W_{WR-12}=3.098$, $H_{WR-12}=1.549$, $L_{1}=10.151$, $L_{2}=1.323$, $L_{3}=9.54$*, $W_{1}=0.34$, $W_{P}=0.228$, $W_{M}=0.127$, D=0.407, G=0.402, $D_{Flange}=19.1$, Thickness=0.127 (all dimensions in mm)).

**Figure 4 sensors-25-05492-f004:**
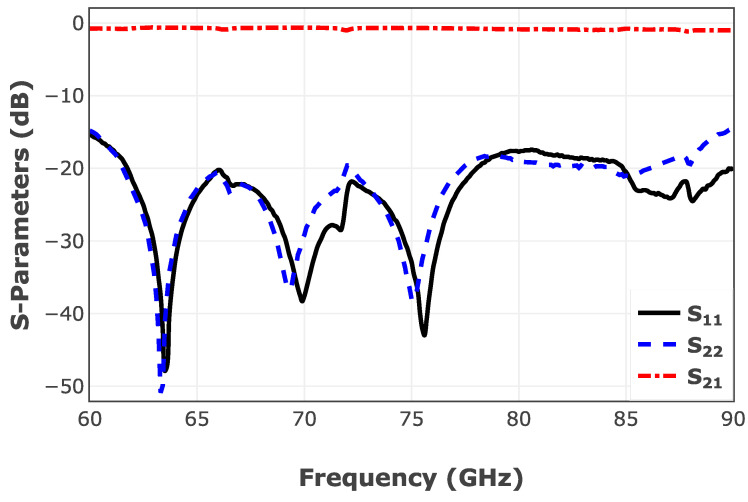
Simulation results of the designed E-Band microstrip to WR-12 transition.

**Figure 5 sensors-25-05492-f005:**
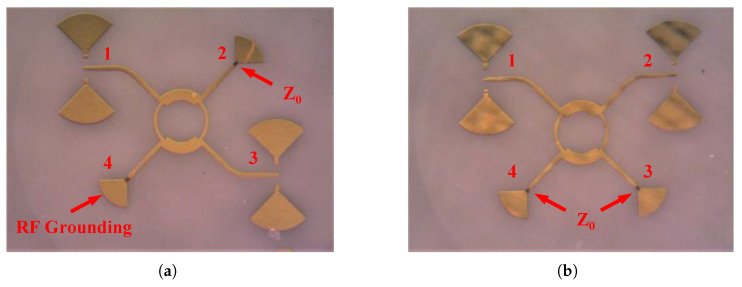
A microphotograph of the proposed E-Band $90^{\circ }$ hybrid coupler; coupling measurement layouts for (**a**) $S_{13}$ and (**b**) $S_{12}$.

**Figure 6 sensors-25-05492-f006:**
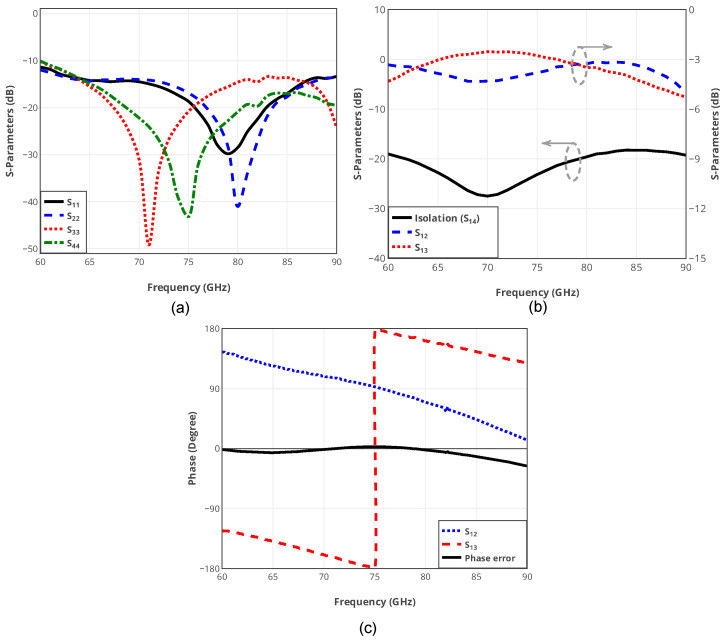
Measurement results of the designed wide-band $90^{\circ }$ hybrid coupler: (**a**) return loss, (**b**) $S_{12}$, $S_{13}$, and $S_{14}$, (**c**) Phase of $S_{12}$, $S_{13}$.

**Figure 7 sensors-25-05492-f007:**
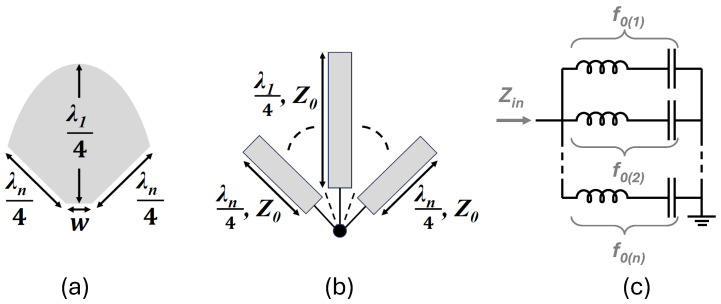
(**a**) The layout of the proposed parabolic open stub; (**b**) its equivalent layout with rectangular open stubs with different lengths; and (**c**) its equivalent circuit.

**Figure 8 sensors-25-05492-f008:**
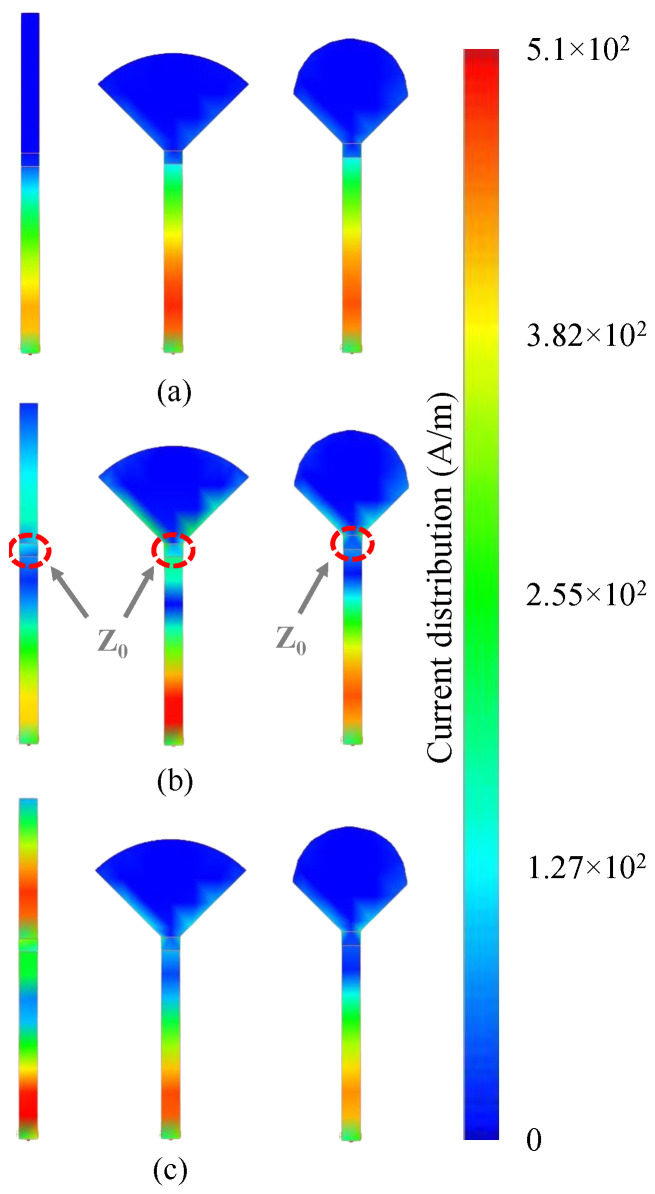
Simulated current distribution of a rectangular, radial, and parabolic open stub at (**a**) 60, (**b**) 75, and (**c**) 90 GHz.

**Figure 9 sensors-25-05492-f009:**
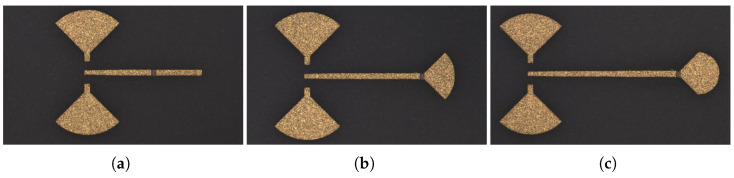
A microphotograph of the designed E-band termination with (**a**) rectangular, (**b**) radial, and (**c**) parabolic open stub as the RF grounding.

**Figure 10 sensors-25-05492-f010:**
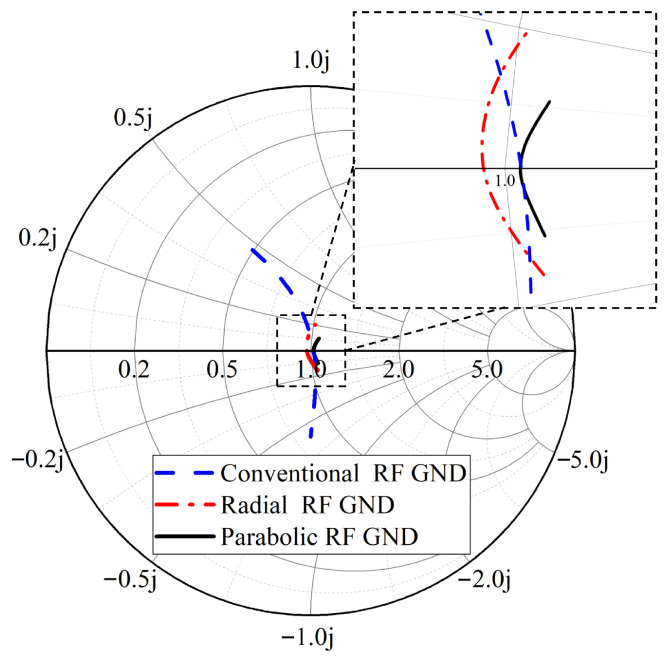
Measurement results of the designed termination using conventional, radial, and parabolic open stubs over the frequency band of 60 to 90 GHz.

**Figure 11 sensors-25-05492-f011:**
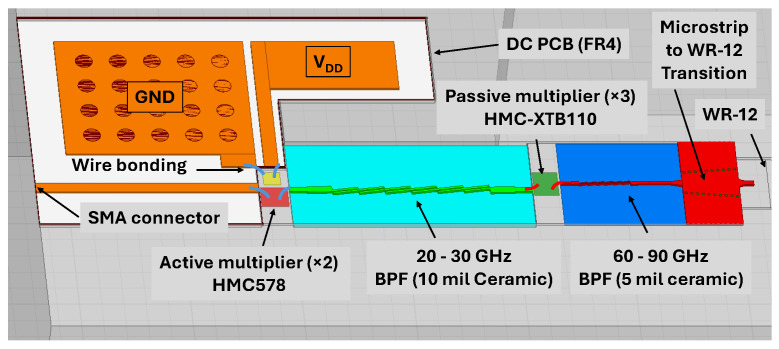
The designed LO chain with the input frequency range of 10 to 15 GHz, and output frequency range of 60 to 90 GHz.

**Figure 12 sensors-25-05492-f012:**
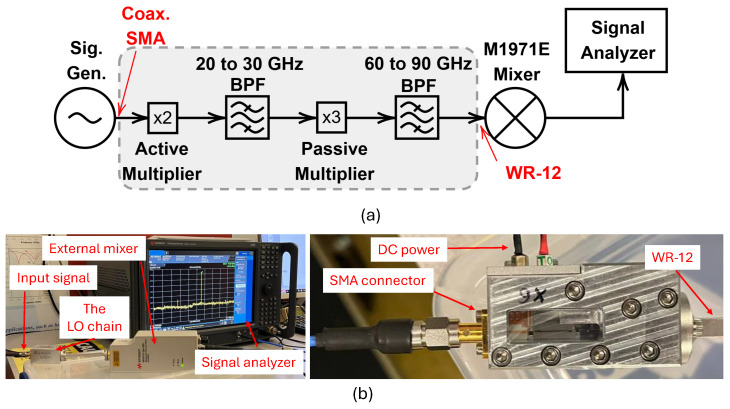
(**a**) The schematic of the measurement setup and (**b**) the designed LO chain under measurement.

**Figure 13 sensors-25-05492-f013:**
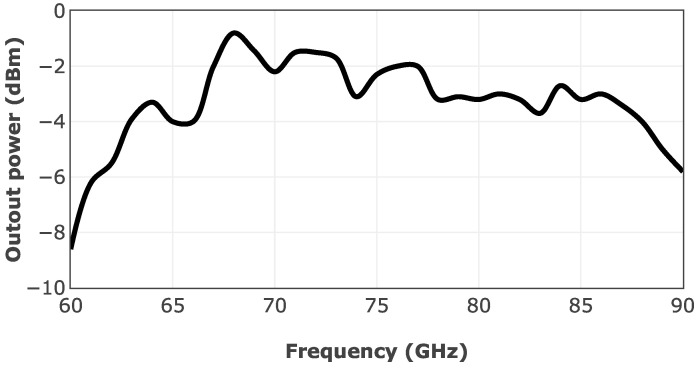
Measurement results of the output power of the designed E-band LO chain vs. output frequency.

**Figure 14 sensors-25-05492-f014:**
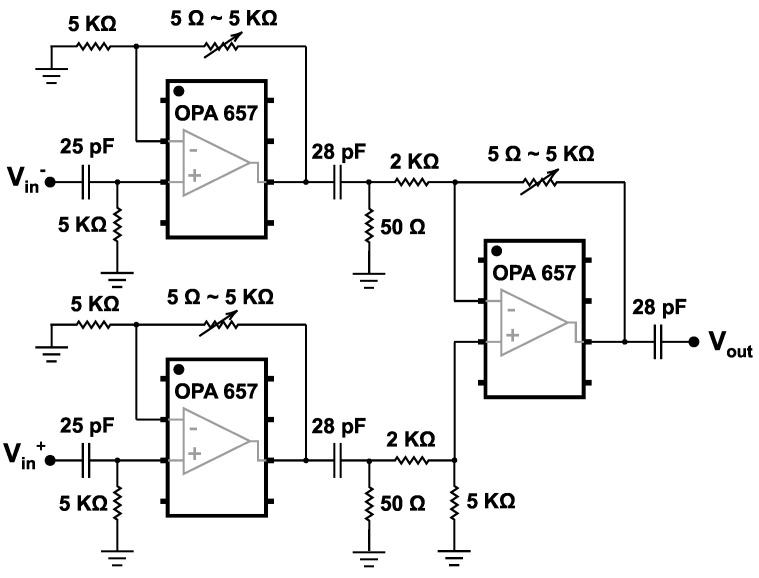
The schematic of the designed differential variable-gain TIA.

**Figure 15 sensors-25-05492-f015:**
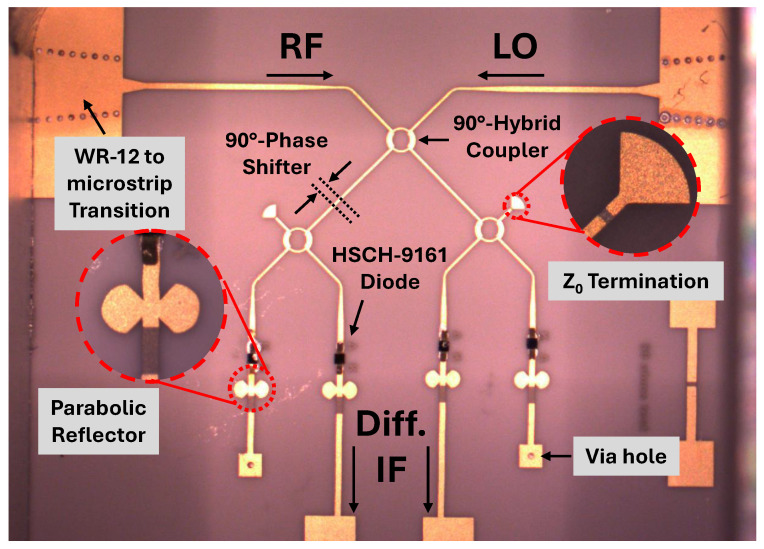
The microphotograph of the designed mixer.

**Figure 16 sensors-25-05492-f016:**
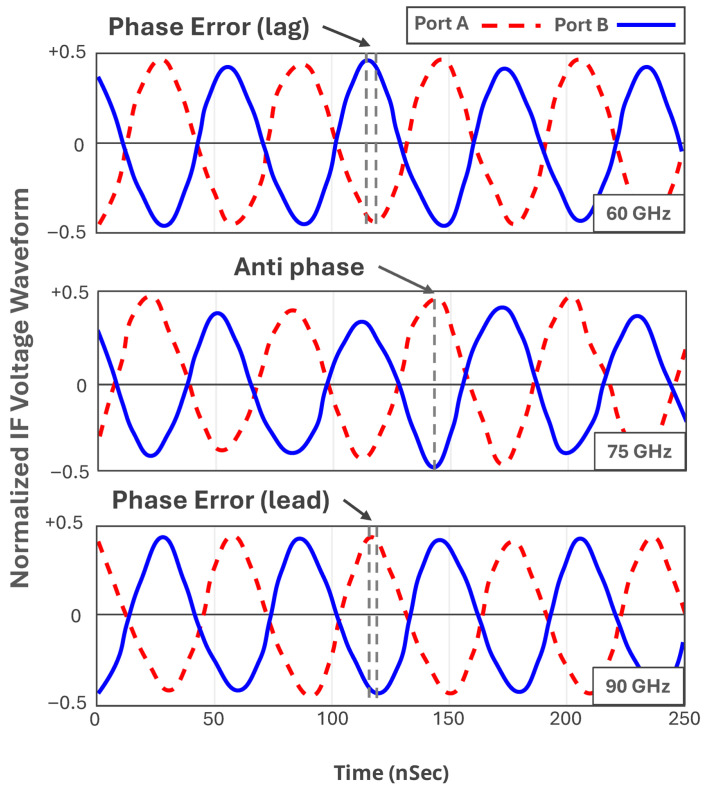
Normalized IF voltage waveforms at the ports A and B of the mixer, while RF is set at 60, 75, and 90 GHz and IF is set to be at 16 MHz.

**Figure 17 sensors-25-05492-f017:**
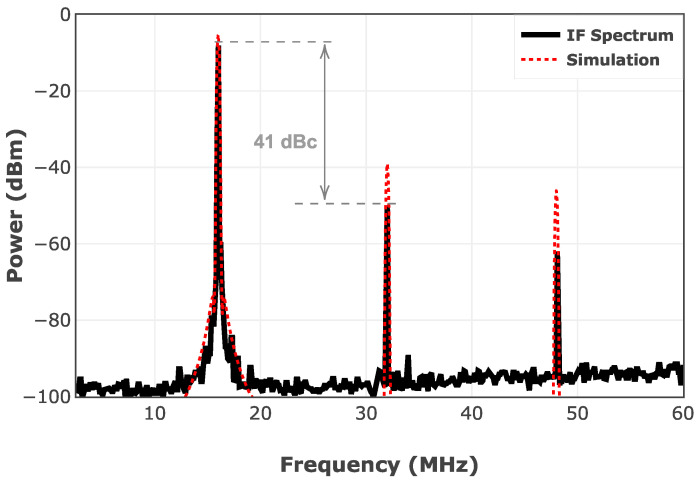
Measured versus simulated IF spectrum of the proposed mixer: *RF* @ 75 GHz, and *IF* @ 16 MHz.

**Figure 18 sensors-25-05492-f018:**
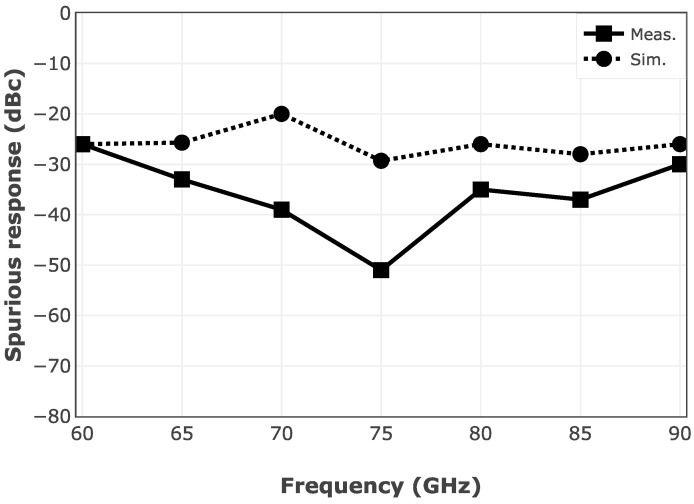
Simulation versus measurement results of the spurious response of the designed mixer.

**Figure 19 sensors-25-05492-f019:**
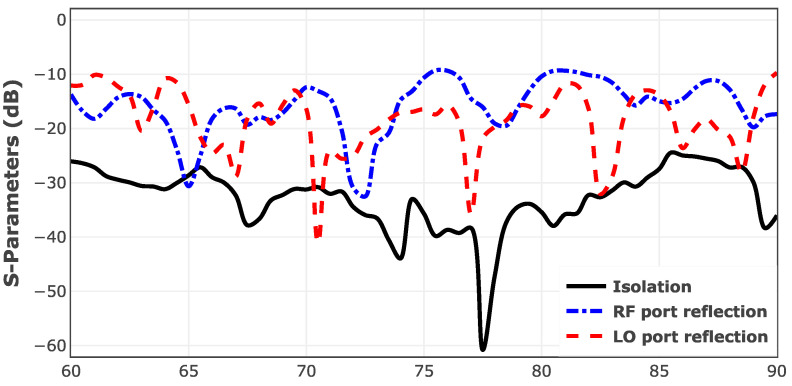
Two-port measurement results of the designed mixer between the LO and RF ports.

**Figure 20 sensors-25-05492-f020:**
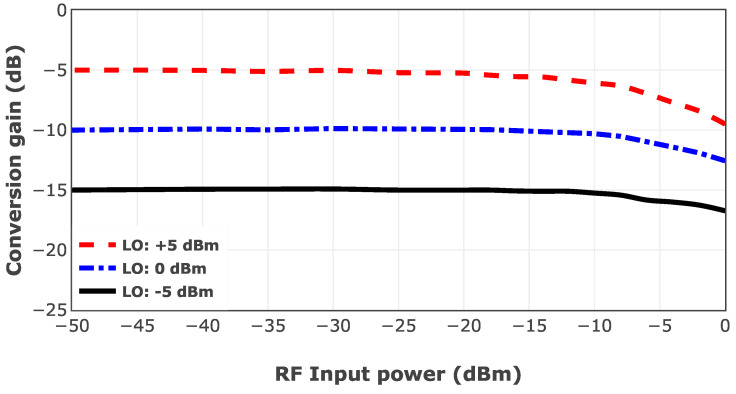
Measured conversion gain of the mixer at 75 GHz with LO power levels of −5, 0, and +5 dBm.

**Figure 21 sensors-25-05492-f021:**
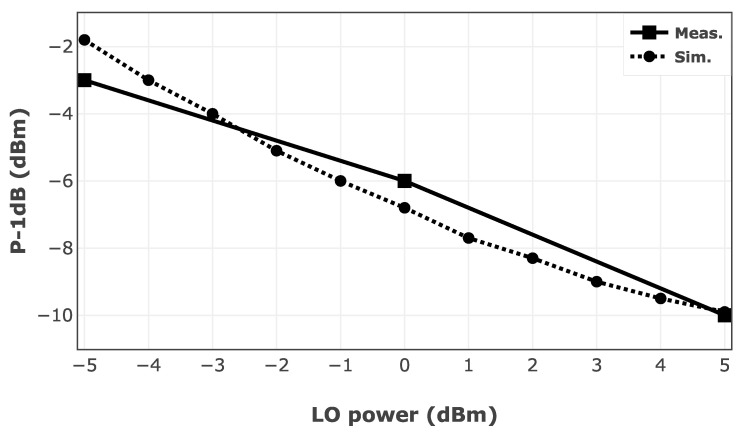
Measurement results of the P-1 dB of the proposed mixer vs. LO powers at RF frequency of 75 GHz and IF frequency of 16 MHz.

**Figure 22 sensors-25-05492-f022:**
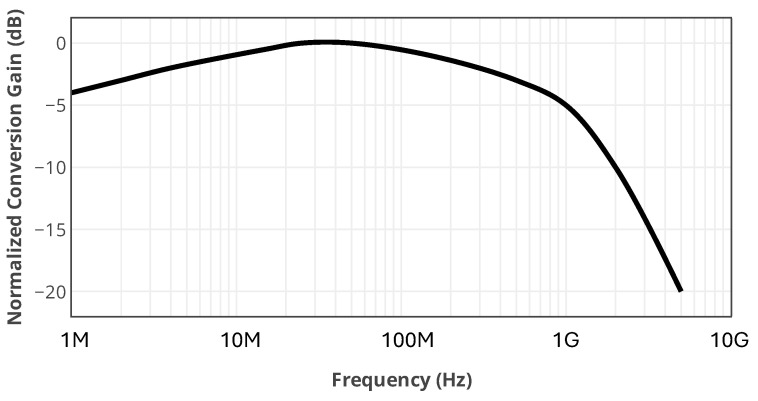
Measurement results of the normalized conversion gain of the mixer versus IF frequency.

**Figure 23 sensors-25-05492-f023:**
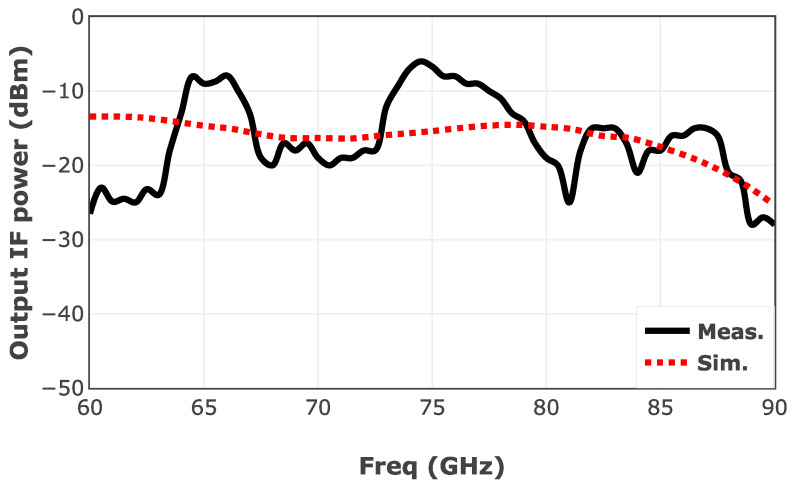
Measured IF power of the designed down-converter versus RF frequency.

**Table 1 sensors-25-05492-t001:** Performance summary and comparison to similar works.

Reference	[41]	[42]	[43]	[44]	This Work
**Technology**	PCB	GaAs	CMOS (65 nm)	SiGe (90 nm)	MHMIC
**Return loss (dB)**	N/A	>12	>7	>5	>10
**Conv. gain (dB)**	−10	∼−5	+4.5	+2	−5
**RF BW (GHz)**	27	35	8	38	30
**Freq. band**	K/Ka	W	K/Ka	D	E
**P-1 dB (dBm)**	N/A	N/A	−6	N/A	−6
**Spurious (dBc)**	N/A	N/A	N/A	N/A	−41
**Variable gain**	No	No	No	No	Yes

## Data Availability

Data are contained within the article.

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
