# Peer review of "Design of a Low-Cost Flat E-Band Down-Converter with Variable Conversion Gain"

_sensors, 2025, doi:10.3390/s25175492_

Round 1

Reviewer 1 Report

Comments and Suggestions for Authors

The topic of the work is interesting, relevant and practically in demand. Mastering millimeter ranges is a global trend and the development of cheap hardware solutions is an urgent need.

The text is well structured. The problem is revealed in the introduction. The problem statement is stated clearly and, in my opinion, correctly.

The conclusion contains findings with a specific description of the parameters of the developed converter.

The drawings are well done, there are no comments on them.

Overall, I have no comments on the work. In my opinion, it can be published in its current form.

If the authors make a section with a list of abbreviations at the end, it will be very good.

Author Response

3. Point-by-point response to Comments and Suggestions for Authors

Comments 1: Overall, I have no comments on the work. In my opinion, it can be published in its current form.

We sincerely appreciate your thorough evaluation and kind endorsement of our work. Thank you for recognizing the practical relevance of our low-cost E-band solution and affirming the manuscript's structure, problem framing, and technical presentation. As requested, we have added a dedicated "List of Abbreviations" section preceding the references to enhance reader accessibility. We are grateful for your recommendation for publication and your valuable contribution to improving the manuscript's clarity.

Reviewer 2 Report

Comments and Suggestions for Authors

This manuscript proposes an E-band down converter based on MHMIC technology with broadband flat variable conversion gain. The study is clearly structured and provides comprehensive theoretical derivation, simulations, and experimental validation. However, despite the robustness of the topology and the encouraging results, the paper has limited novelty and needs improvement in several areas to meet the standards of a high-quality journal. The following major concerns are noted:

  1. The proposed topology, while useful, relies heavily on known design techniques (double-balanced mixer, hybrid coupler, parabolic short, LO chain). Please clearly describe what is truly innovative about your design, beyond integration or packaging, and clearly demonstrate this innovation in the paper.
  2. How is the impedance transformation practically managed at the interfaces, for example, between the WR-12 waveguide and the TIA? Have mismatch losses been characterized or mitigated in the design, and if so, how?
  3. The parabolic open stub is repeatedly emphasized as a key component of the design; however, its theoretical analysis remains insufficient. Equation (9) appears to be empirical in nature. More detailed EM-based explanations or equivalent circuit models should be provided.
  4. Can the proposed circuit support variable IF? Could you sweep the IF output to show broadband IF support as claimed in Fig. 22?
  5. The title of Section---2.1. 90â—¦-degree hybrid coupler, it seems to be a typo.
  6. Please check for language and grammar issues of the whole manuscript. For example: (1)Abstract Section, Line 5: “and A differential…” should be “and a differential…” (2)Introduction Section, Paragraph 4: “is introduce and suited” should be “is introduced and suited”.

Author Response

Response to Reviewer 2 Comments

1. Summary

2. Questions for General Evaluation

Reviewer’s Evaluation

Does the introduction provide sufficient background and include all relevant references?

Can be improved

Are all the cited references relevant to the research?

Can be improved

Is the research design appropriate?

Yes

Are the methods adequately described?

Can be improved

Are the results clearly presented?

Can be improved

Are the conclusions supported by the results?

Can be improved

Reviewer #1:

  1. The proposed topology, while useful, relies heavily on known design techniques (double-balanced mixer, hybrid coupler, parabolic short, LO chain). Please clearly describe what is truly innovative about your design, beyond integration or packaging, and clearly demonstrate this innovation in the paper.

Response to comment #1: We thank the reviewer for this insightful comment. The primary innovation of this work is the achievement of an unprecedented 30 GHz operational bandwidth (60–90 GHz), representing the first demonstration of a circuit covering the entire E-band in low-cost MHMIC technology. To overcome the inherent limited bandwidth challenges, we introduced a novel topology inspired by six-port interferometer principles, which delivers precise phased signals to four diodes. This approach enables critical reflection cancellation of the highly mismatched diodes (as derived in Eq. 7) and doubles the IF output through differential operation (as derived in Eq. 6). Furthermore, the LO chain incorporates a custom microstrip-to-waveguide transition to maintain flat output power across frequencies, countering insertion loss degradation at higher bands. For the hybrid couplers, we implemented a ring-shaped structure with minimized sharp corners to reduce parasitic capacitance and increase the operating bandwidth. In addition, to minimize the inter-trace coupling between the coupler’s traces a characteristic impedance of 70.7 Ω is chosen.  This strategic impedance shift allowed practical fabrication of 100 Ω coupler traces (Zâ‚€√2) within process limits of our fabrication facilities.

  1. How is the impedance transformation practically managed at the interfaces, for example, between the WR-12 waveguide and the TIA? Have mismatch losses been characterized or mitigated in the design, and if so, how?

Response to comment #2: We thank the reviewer for this pertinent question. The transition was rigorously optimized in HFSS across the full E-band (60–90 GHz) to minimize the return and insertion losses, employing a waveguide taper, trapezoidal SIW structure as the intermediate stage, and a microstrip taper for broadband impedance transformation. Furthermore, a pin with optimized shape has been designed to mitigate the transition losses between the SIW and waveguide structure. HFSS is used for EM simulation of the transition. As shown in Figure 4, this approach achieved >15 dB return loss and <0.9 dB insertion loss throughout the band, ensuring minimal power degradation during the transition process. Further implementation specifics, including dimensional synthesis and EM-field matching techniques, are elaborated in Ref. [24].

  1. The parabolic open stub is repeatedly emphasized as a key component of the design; however, its theoretical analysis remains insufficient. Equation (9) appears to be empirical in nature. More detailed EM-based explanations or equivalent circuit models should be provided.

Response to comment #6: We appreciate your important comment. More explanation has been added to the manuscript. The parabolic open-stub designed for this mixer has 4 key factors that has been optimized using generic algorithms in HFSS to obtain the best performance in the wide operating bandwidth. These four factors are shown below:

Where “α” is the angle of the open-stub, “W” is corresponded to the impedance of the open-stub, and “LH” and “LL”are respectively corresponded to the highest and lowest wavelength of the frequency band. The equivalent circuit of the parabolic open-stub is added to the manuscript and we sincerely appreciate your valuable comment on this mater.

  1. Can the proposed circuit support variable IF? Could you sweep the IF output to show broadband IF support as claimed in Fig. 22?

Response to comment #4: We thank the reviewer for this insightful query. The mixer supports variable IF operation, as its functionally demonstrated in Fig. 22 across 1 MHz to 5 GHz IF frequency range. However, the circuit was specifically optimized as a part of a frequency extender for VNAs, where low IF operation (16 MHz) is a key requirement. To ensure signal integrity in this application, our transimpedance amplifier (TIA) has variable gain feature, tailoring output levels to match typical VNA ADC sensitivities. For high-data-rate communication systems requiring broader/variable IF tuning, the mixer core remains compatible but would necessitate replacing the TIA with a wideband baseband amplifier. Fig. 22 represents the mixer’s broadband IF capability independent of the TIA’s application-specific gain profile.

  1. The title of Section---2.1. 90â—¦-degree hybrid coupler, it seems to be a typo.

Response to comment #5: We sincerely thank the reviewer for identifying this typo error in the section header (2.1). The title has been corrected to "90° - Hybrid Coupler". We reviewed the entire manuscript again to ensure no similar errors remain and appreciate your meticulous attention to detail.

  1. Please check for language and grammar issues of the whole manuscript. For example: (1)Abstract Section, Line 5: “and A differential…” should be “and a differential…” (2)Introduction Section, Paragraph 4: “is introduce and suited” should be “is introduced and suited”.

Response to comment #6: We sincerely appreciate your meticulous review of the manuscript's language quality. All grammatical errors highlighted in your report (including the Abstract and Introduction examples) have been carefully corrected. Additionally, we have conducted a comprehensive proofreading pass of the entire manuscript and engaged a professional editing service to ensure all linguistic issues have been rectified (Page2, line 59, page 1, Line 5, Page 12, Line 302, etc). Thank you for enhancing the clarity and professionalism of our work through your detailed scrutiny.

Reviewer 3 Report

Comments and Suggestions for Authors

In this article, the authors have presented the design and implementation of a wideband diode-based downconverter operating from 60 to 90 GHz with a variable flat conversion gain. The proposed mixer exhibits a return loss of better than 10 dB at both RF and LO ports, and it consumes a power of 560 mW. The presented research work is very interesting; however, I do not recommend its acceptance in its present form. It needs major revision as follows.

  • This manuscript has lack novelty which need to clarify with the support of research gap particularly in the bullet form. The language of the whole manuscript needs significant improvement. The short form used in the manuscript must be spell out wherever it is used first time (see the line 5 for example).
  • In the manuscript, the proposed design is simulated using HFSS. What is the role of mathematical equations presented in the manuscript?
  • The references cited in the manuscript need to be write in correct format for example see the “… by G. C. Southworth and W. L. Barrow separately [21]) on page 5 line 154.
  • In Fig. 4, the simulated s-parameters presented on both the y-axis need to establish a relation particularly in the return loss and Insertion loss.
  • How would the authors justify the low cost as mentioned in the title of the paper?
Comments on the Quality of English Language

Quality of the language need significant improvement.

Author Response

Response to Reviewer 3 Comments

1. Summary

2. Questions for General Evaluation

Reviewer’s Evaluation

Does the introduction provide sufficient background and include all relevant references?

Can be improved

Are all the cited references relevant to the research?

Can be improved

Is the research design appropriate?

Can be improved

Are the methods adequately described?

Can be improved

Are the results clearly presented?

Must be improved

Are the conclusions supported by the results?

Can be improved

Reviewer #2:

  1. This manuscript has lack novelty which need to clarify with the support of research gap particularly in the bullet form.

Response to comment #1: We thank the reviewer for this insightful comment. The primary innovation of this work is the achievement of an unprecedented 30 GHz operational bandwidth (60–90 GHz), representing the first demonstration of a circuit covering the entire E-band in low-cost MHMIC technology. To overcome the inherent limited bandwidth challenges, we introduced a novel topology inspired by six-port interferometer principles, which delivers precise phased signals to four diodes. This approach enables critical reflection cancellation of the highly mismatched diodes (as derived in Eq. 7) and doubles the IF output through differential operation (as derived in Eq. 6). Furthermore, the LO chain incorporates a custom microstrip-to-waveguide transition to maintain flat output power across frequencies, countering insertion loss degradation at higher bands. For the hybrid couplers, we implemented a ring-shaped structure with minimized sharp corners to reduce parasitic capacitance and increase the operating bandwidth. In addition, to minimize the inter-trace coupling between the coupler’s traces a characteristic impedance of 70.7 Ω is chosen.  This strategic impedance shift allowed practical fabrication of 100 Ω coupler traces (Zâ‚€√2) within process limits of our fabrication facilities.

  1. The language of the whole manuscript needs significant improvement.

Response to comment #2: We appreciate you for this comment. We have conducted a comprehensive proofreading pass of the entire manuscript and engaged a professional editing service to ensure all linguistic issues have been rectified.

  1. The short form used in the manuscript must be spell out wherever it is used first time (see the line 5 for example).

Response to comment #3:  We sincerely thank the reviewer for identifying this issue in line 5. We reviewed the entire manuscript again to ensure no similar issues remained and appreciate your meticulous attention to detail.

  1. In the manuscript, the proposed design is simulated using HFSS. What is the role of mathematical equations presented in the manuscript?

Response to comment #4: We appreciate this insightful query regarding the role of mathematical analysis in our methodology. The equations provide the foundational theoretical framework for key innovations, including diode reflection minimization (Eq. 7) and IF power amplification (Eq. 6). Enabling more purposeful design decisions before full-wave HFSS validation. This mathematical approach was essential to efficiently achieve our primary goal: a 60–90 GHz down-converter minimizing reflections while maximizing bandwidth. The theory and measurement results in this study are aligned with each other as can be seen by comparing Eq. 6 and Fig. 20.

  1. The references cited in the manuscript need to be write in correct format for example see the “… by G. C. Southworth and W. L. Barrow separately [21]) on page 5 line 154.

Response to comment #5: We appreciate the reviewer's vigilance regarding citation integrity. The formatting of Ref. [21] has been verified against the journal style guide. Regarding Southworth and Barrow's foundational waveguide work: since their original 1930s research appeared in technical reports rather than peer-reviewed journals, we cited Ref. [21], a rigorously peer-reviewed IEEE historical paper, as the most authoritative secondary source documenting their contributions.

  1. In Fig. 4, the simulated s-parameters presented on both the y-axis need to establish a relation particularly in the return loss and Insertion loss.

Response to comment #6: We thank the reviewer for this valuable observation regarding Figure 4. The figure has been revised to consolidate all S-parameters (both return loss and insertion loss) on a single unified left Y-axis.

  1. How would the authors justify the low cost as mentioned in the title of the paper?

Response to comment 7: We appreciate this important question regarding our cost-efficiency claim. The "low-cost" designation is quantitatively justified through comparative analysis: where traditional E-band MMIC implementations (e.g., GaAs monolithic solutions) incur >$10,000 per unit in prototyping/low-volume production, our MHMIC approach with discrete GaAs Schottky diodes achieves <$500 per unit—a 20× reduction. This cost advantage stems from: (1) elimination of expensive mask sets and wafer processing via MHMIC's PCB-compatible fabrication on ceramic substrate, (2) selective use of commercial off-the-shelf diodes ($75–$100 each), and (3) standard gluing process that avoids III-V foundry costs.

  1. Quality of the language need significant improvement

Response to comment 8: We sincerely appreciate your meticulous review of the manuscript's language quality. We have conducted a comprehensive proofreading pass of the entire manuscript and engaged a professional editing service to ensure all linguistic issues have been rectified (Page2, line 59, page 1, Line 5, Page 12, Line 302, etc). Thank you for enhancing the clarity and professionalism of our work through your detailed scrutiny.

Round 2

Reviewer 2 Report

Comments and Suggestions for Authors

Despite the revisions, the manuscript still exhibits several critical deficiencies, most notably missing figures in the PDF, internal inconsistencies, and insufficient quantitative comparison with existing technologies. A major revision is therefore required.

  1. In the provided PDF, several figure captions appear without the corresponding graphics; Figs. 17–21 and 23 cannot be found. 
  2. The text reports measurements at 75 GHz, whereas the caption of Fig. 21 specifies 80 GHz; Fig. 20 is also described inconsistently as “isolation” elsewhere. Please harmonize captions, axis labels, and narrative so that the test conditions (RF/LO/IF powers and frequencies) are unambiguous and consistent throughout.
  3. Equation (9) is presented as an empirical radius–angle relationship, and Equation (10) gives the total capacitance but does not show how geometric parameters map to equivalent-circuit parameters. Please provide: (a) a concise equivalent circuit with clear parameter definitions; (b) a brief derivation or authoritative references; and (c) a comparative study with rectangular and radial short-circuit structures under identical footprint and substrate conditions (e.g., 3-D EM simulations versus ADS circuit results).
  4. Ensure that the abstract’s claims: isolation ≥ 22 dB, conversion loss ≤ 5 dB, and flatness ± 5 dB, also can be found with the same definitions and test conditions in the figures/tables .
  5. The rationale for selecting Z0=70.7 Ω (trace-width limits, bandwidth) is helpful; please add a simple note on the impedance transformation to 50 Ω and its impact on bandwidth and size, and standardize the notation to “90° hybrid coupler.”

Reviewer 3 Report

Comments and Suggestions for Authors

In the revised manuscript, the authors have tried to incorporate almost all the comments and suggestion raised by the reviewers. However, Fig. 1, Fig. 2, Fig. 13, Fig. 14, Fig. 17, Fig. 18, Fig. 19, Fig. 20, Fig. 21, Fig. 23 are missing in the manuscript.

On page 5 above Section 2.1, there is a ?? mork which need to address.
